# Molecular Probes to Evaluate the Synthesis and Production Potential of an Odorous Compound (2-methylisoborneol) in Cyanobacteria

**DOI:** 10.3390/ijerph17061933

**Published:** 2020-03-16

**Authors:** Keonhee Kim, Youngdae Yoon, Hyukjin Cho, Soon-Jin Hwang

**Affiliations:** 1Human and Eco-Care Center, Department of Environmental Health Science, Konkuk University, Seoul 05029, Korea; skyopera@konkuk.ac.kr (K.K.); yyoon21@gmail.com (Y.Y.); 2Department of Environmental Health Science, Konkuk University, Seoul 05029, Korea; 3Hangang River Regional Division, Department of Water Resources Management, K-Water, Gwacheon 13841, Korea; hyukjin@kwater.or.kr

**Keywords:** 2-Methylisoborneol (2-MIB), cyanobacteria, molecular probes, *mibC*, real-time PCR

## Abstract

The volatile metabolite, 2-Methylisoborneol (2-MIB) produced by cyanobacterial species, causes odor and taste problems in freshwater systems. However, simple identification of cyanobacteria that produce such off-flavors may be insufficient to establish the causal agent of off-flavor-related problems as the production-related genes are often strain-specific. Here, we designed a set of primers for detecting and quantifying 2-MIB-synthesizing cyanobacteria based on *mibC* gene sequences (encoding 2-MIB synthesis-catalyzing monoterpene cyclase) from various Oscillatoriales and Synechococcales cyanobacterial strains deposited in GenBank. Cyanobacterial cells and environmental DNA and RNA were collected from both the water column and sediment of a eutrophic stream (the Gong-ji Stream, Chuncheon, South Korea), which has a high 2-MIB concentration. Primer sets mibC196 and mibC300 showed universality to *mibC* in the Synechococcales and Oscillatoriales strains; the mibC132 primer showed high specificity for *Pseudanabaena* and *Planktothricoides mibC*. Our mibC primers showed excellent amplification efficiency (100–102%) and high correlation among related variables (2-MIB concentration with water RNA r = 689, *p* < 0.01; sediment DNA r = 0.794, *p* < 0.01; and water DNA r = 0.644, *p* < 0.05; cyanobacteria cell density with water RNA and DNA r = 0.995, *p* < 0.01). These primers offer an efficient tool for identifying cyanobacterial strains possessing *mibC* genes (and thus 2-MIB-producing potential) and for evaluating *mibC* gene expression as an early warning of massive cyanobacterial occurrence.

## 1. Introduction

Naturally occurring 2-methylisoborneol (2-MIB) and geosmin produced by cyanobacteria and actinomyces are regarded as a worldwide problem in freshwater systems because of their muddy or earthy odorous properties [1,2]. Contamination of drinking water resources is of particular concern, motivating intensive studies to identify new methods for efficiently detecting and removing these compounds [1,3,4].

Cyanobacteria are now recognized as the major producer of the unpleasant odorous 2-MIB; thus, the odor is more severe during cyanobacterial blooms. Prior studies indicated that blooms of cyanobacteria coincided with the occurrence of odor substances in various freshwater systems [5,6], including Huajiang Lake in Mongol, where the maximum concentration of 2-MIB reached 102 ng/L during cyanobacterial blooms [7]. The major cyanobacterial genus in the North Han River in Korea has shifted from *Anabaena* to *Pseudanabaena* since 2013, accompanied by an increase in the occurrence and concentration of 2-MIB with the maximum concentration of 74.5 ng/L recorded in 2014 [8,9]. However, a direct causal relationship between the blooms of cyanobacteria as a whole and the generation of odor substances is unclear because the cyanobacterial community in freshwater systems is composed of a wide variety of species related to odor production, making it difficult to specify which species are responsible for the production of the odor-causing compounds [10,11].

There has been increasing research in the development of methods related to 2-MIB detection in recent decades as this odorous compound has become increasingly problematic for drinking water resources [12]. Various research groups have proposed diverse approaches for detecting odorous compounds. In particular, because even rarely occurring species of cyanobacteria in certain freshwater systems can produce odorous substances, it is necessary to apply highly sensitive and specific methods. Molecular biological approaches such as polymerase chain reaction (PCR), are very relevant in this regard [13,14]. The biochemical mechanisms and genes involved in 2-MIB synthesis in various cyanobacteria genera, including *Pseudanabaena*, *Planktothricoides*, *Lyngbya*, and *Oscillatoria* (*Limnothrix*), have been revealed; therefore, cyanobacterial strains capable of synthesizing 2-MIB can now be detected by PCR analysis using sequence-specific primers for the *mibC* gene encoding a monoterpene cyclase, which is the main enzyme involved in 2-MIB synthesis [12,15,16,17,18]. Using a PCR approach, the odor compound-producing potential in an Australian eutrophic reservoir was found to be positively correlated with the concentrations of geosmin compounds but not to the cell density of cyanobacteria known to produce the odor compounds [19]. 

To facilitate more selective and specific detection of cyanobacteria possessing the *mibC* gene, in this study, we constructed diverse sets of primers targeting *mibC* among species of Synechococcales and Oscillatoriales and investigated the 2-MIB production capability in 14 different cyanobacterial strains using these primer sets. To improve the specificity and efficiency of the primers, various annealing temperatures were tested in PCR amplification, and detectability of 2-MIB was assessed through application of the target gene to field samples.

## 2. Materials and Methods

### 2.1. Preparation of Cyanobacterial Strains and Field Samples

The cyanobacterial strains used in this study were *Pseudanabaena galeata* (NIES-512), *Anabaena smithii* (NIES-824), and *Anabaena cylindrica* (NIES-19) obtained from the National Institute of Environment Science (NIES, Japan), and *Anabaena variabilis* (AG10064) obtained from the Korea Collection for Type Cultures (KTCC). The other strains, including *Microcystis aeruginosa*, *Dolichospermum circinale* straight type (ST), *D. circinale* coiled type (CT), *Planktothricoides raciborskii* strain NO, *P. raciborskii* strain PO, *Pseudanabaena mucicola*, *Oscillatoria redekei* (*Limnothrix redekei*), *Lyngbya murrayi*, *Phormidium ambiguum,* and *Leptolyngbya boryana*, were isolated from the North Han River and identified based on DNA sequence in previous studies [8,20,21]. All cyanobacterial strains were inoculated in BG-11 medium [22] and cultured in a shaking incubator (VS-3125Qi, Vison, Korea) at 25 °C and a light intensity of 60 μmolE·m^−2^·s^−1^ (light:dark = 14:10 h). The cyanobacterial strains were grown for over three months and the medium was replaced every month. During medium changes, the dead cells that settled at the bottom of the flask and paled into yellow-green color were removed along with the medium.

Field samples were obtained from a site with extensive production of 2-MIB by cyanobacteria that coincided with a cyanobacterial bloom between 2015 and February 2016 [8]. The cyanobacterial community in both the water column and sediment was collected downstream of the Gong-ji Stream (Appendix A) (N 37°52′33.4″, E 127°42′23.1″) using a Vandorn sampler (Horizontal water sampler, iStech, Korea) and grab sampler (Peterson grab, QT Technology, Korea), respectively, monthly from February 2015 to February 2016. The sampling site constitutes the region where the Gong-ji Stream merges with the Uiam reservoir, a large reservoir located in the mainstream of the North Han River. The surface water sample and the sediment sample were transferred to a 1 L clear PET bottle and 100 mL dark bottle, respectively. The field samples were stored at 4 °C in the dark before pretreatment. Sample pretreatment was performed within 12 h. The water sample was filtered using a polycarbonate filter paper (ø25 mm, pore: 1 μm, Whatman, USA) and placed in a 2-mL microtube before DNA extraction. The sediment sample (0.8 g) was transferred to a silica bead tube. Both the polycarbonate filter-filtered water sample and the sediment sample within the silica bead tube were stored at −80 °C until analysis.

### 2.2. Genomic DNA and RNA Extraction

For genomic DNA extraction, cyanobacterial strains (*P. galeata*, *P. mucicola*, *O. redekei* [*L. redekei*], and *P. raciborskii* NO) attached to the bottom of flasks were collected using a scraper, whereas the other strains (*M. aeruginosa*, *D. circinale* ST, *D. circinale* CT, *A. variabilis*, *A. smithii*, *A. cylindrica*, *L. murrayi*, *P. ambiguum*, *L. boryana*, and *P. raciborskii* strain PO) were harvested by filtration of 10 mL of culture using a polycarbonate filter (ø25 mm, pore: 1 μm, Whatman, USA). The genomic DNA of the algal strain was extracted using a DNA extraction kit for soil (Macherey-Nagal, Germany) according to the manufacturer’s instructions, after cell lysis via the physical bead beating method [23]. Both water column and sediment samples were extracted DNA using the same method as for algal strain DNA extraction.

The environmental RNA collected from the water column of the Gong-ji Stream was extracted with Trizol using an RNA purification kit (Hybrid-R^TM^ RNA purification kit, Geneall Co., Korea), according to the manufacturer’s instructions, after cell lysis using the Trizol-based Riboex solution.

The concentration of the DNA and RNA was determined using a Nanodrop system (Thermo Fisher Scientific, USA). 

### 2.3. Sequence Analysis of mibC Genes and Primer Design

The monoterpene cyclase gene *mibC* was chosen as a target for evaluation of 2-MIB production potential given its essential enzymatic role in 2-MIB synthesis (Figure 1) [18]. To specifically detect *mibC* in a wide variety of cyanobacterial strains, sequences from diverse bacteria, including *Pseudanabaena* sp. NIVA-CYA 111 (HQ630887), *Pseudanabaena* sp. dph 15 (HQ830028), *Pseudanabaena galeata* (AB826230.1), *Pseudanabaena limnetica* isolated from Castaic Lake (HQ630883), *Oscillatoria* sp. 327/2 (KJ658377), *Oscillatoria limosa* (AB826230.1), *Planktothricoides* sp. 328.2 (KJ658378), *Planktothricoides raciborskii* CHAB3331(339892252), *Leptolyngbya* sp. A2 (KP013063.1), and *Leptolyngbya* sp. (KM013398.1), were obtained from the National Center for Biotechnology Information (NCBI) GenBank database and analyzed using the Clustal W algorithm of MEGA 6.0 [24,25]. Nucleotide sequences were integrated based on *mibC* gene sequence alignment using the MEGA program. Differences in nucleotides between *mibC* sequences were integrated by the mixed base. We excluded mixed bases with three to four nucleotides to avoid obtaining an unintended PCR product. The mibC primer sequence, which was based on the integrated nucleotide sequences, was designed by the PrimerQuest Tool program (IDT Co., USA) [26]. The melting temperature of each primer was calculated by the PrimerQuest Tool program; then, the annealing temperature in PCR amplification was adjusted by ±5 °C with reference to the calculated melting temperature. The conserved regions identified from sequence alignment were selected, and 20–26-mer primer sets were designed to produce 132–719 bp amplicons. The primers were used for qualitative screening and quantitative analysis based on the length of the amplicons. Primers such as mibC117, mibC127, and mibC132 producing PCR products shorter than 150 bp were used for real-time PCR, and the mibC300, mibC564, and mibC719 primers were used for screening. mibC196 and mibC200 primers producing amplicons with a length of 150–200 bp were used for both screening and real-time PCR.

### 2.4. Detection of mibC and Field Application

To verify the capability of the designed primers for *mibC* detection, PCR was performed using genomic DNA extracted from 10 cyanobacterial strains as templates. *P. galeata* NIES-512, which is known to produce 2-MIB [27,28], was used as a positive control strain, and *M. aeruginosa*, *D. circinale* ST, *D. circinale* CT, *A. variabilis*, *A. smithii*, *A. cylindrica*, and *P. raciborskii* strain NO, which were confirmed not to produce 2-MIB by gas-chromatography analysis [8,21,29,30,31], were used as negative controls. Strains *L. redekei*, *L. murrayi*, *P. ambiguum*, *P. mucicola*, *P. raciborskii* strain PO, and *L. boryana* isolated from the North Han River watershed were the test strains for determining the potential to produce 2-MIB. To achieve the specific detection of *mibC*, the experimental conditions for PCR were optimized by testing different annealing temperatures for each primer set. In brief, 35 cycles of PCR amplification were performed with 25 s of denaturation at 95 °C, 50 s of annealing at diverse temperatures (58 °C–64 °C), and 50 s of extension at 72 °C. The PCR products were analyzed by agarose electrophoresis and the amplicons were confirmed based on the molecular size of the DNA bands. For the field application, the environmental DNA and RNA of the cyanobacterial community extracted from the water column and sediment of the estuary of Gong-ji Stream were used as the template for PCR and analyzed by agarose gel electrophoresis. The electrophoresis results were photographed (350D, Canon Co., Japan) for analysis. The target band obtained on electrophoresis was used for nucleotide sequence analysis after agarose gel cutting. The bands of the agarose gel were cut by a surgical scalpel after approximating each size (ca. 5 mm × 3 mm) by a ruler. From the middle of agarose gel band, about 2 mm of band edge was removed. The PCR product was extracted from the band cut using a gel & PCR purification system (HiGene^TM^, GP104-200, Biofact Co., Korea).

In addition, real-time PCR was performed using the 2× Real-Time PCR Master Mix: including SFCgreen^®^I in mixture (DQ362-40h, Biofact Co. Korea) in a real-time PCR cycler (Rotorgene, Qiagen Co., Germany) to verify the applicability of the designed primers as *mibC* gene detection probes for quantifying RNA expression and the DNA copy number distribution. Each template of DNA and RNA was mixed with polymerase mixture for the real-time PCR analysis. The mixture was blended with all necessary materials, including 2× polymerase master mix (SFCgreen-intercalate type) 10 μL, forward primer (10 pmole/μL) 1 μL, reverse primer (10 pmole/μL) 1 μL, template 5 μL, and nuclease free water 3 μL. Real-time PCR was performed at 94 °C for 15 min, followed by 35 cycles of 15 s of denaturation at 95 °C, 25 s of annealing at the respective melting temperature (Table 1), and 30 s of extension at 72 °C. The SFCgreen fluorescence value was obtained in each cycle of the extension step. The copy number of the *mibC* gene was calculated using Equation (1), which was modified from an equation for calculating the copy number of genes (N_cn_) in environmental samples using real-time PCR external standard data [32]:(1)$$N_{cn}=\frac{C_{T}\times A_{N}}{S_{A}\times MW_{bp}\times 10^{6}}$$
where *C_T_* is the copy number of the target gene in the DNA sample (copy/μg) estimated using an external standard; *A_N_* and *MW_bp_* are Avogadro’s number and the average molecular weight of a base pair, respectively [33]; and *S_A_* is the genome size (bp) of *Pseudanabaena*. 

Correlative relationships were analyzed among *mibC* copy number, cyanobacterial cell density, and concentration of 2-MIB using a statistical program (PASW Statics 18, USA). The 2-MIB concentration was measured according to standard methods for drinking water surveillance [34].

### 2.5. Sequence Analysis of mibC

The single amplicons obtained from PCR were sequenced using an ABI 373 0XL DNA analyzer (Perkin Elmer, USA). A single DNA band was purified using a PCR product purification kit (Bioneer, Korea) and amplified using ABI BigDye Terminator v3.1 Cycle-Sequencing Kit for DNA sequencing. The DNA sequences were further analyzed using the BLAST program provided by the NCBI GenBank database.

### 2.6. Phylogenetic Analysis

The DNA sequence of the *mibC* gene amplified from the field samples was aligned with the reported sequences of *mibC* genes from diverse strains used for sequence alignment with Clustal W, as described above. Phylogenetic analysis of the aligned sequences was performed using the maximum-likelihood method in MEGA 6.0 software [24] to construct a phylogenetic tree. The stability of the phylogenetic tree was evaluated by bootstrap analysis with 1000 replicates [35].

## 3. Results

### 3.1. Applicability of the Designed Primers

To detect the 2-MIB-synthesizing gene *mibC* in cyanobacteria, eight sets of primers were constructed based on the sequence alignment of *mibC* genes from diverse cyanobacteria (Table 1). The average guanine-cytosine (G-C) content of each primer was 52%, ranging from 40% to 60%. The melting point of each primer ranged from 60 °C to 64 °C, and the average calculated dissociation temperature of the forward and reverse primers was used as the annealing temperature for PCR. We verified the target amplicons and non-specific amplicons from the general PCR in nine cyanobacteria control strains with the eight sets of primers. As shown in Figure 2, seven sets of primers (i.e., all primers except for mibC127) showed the expected sizes of amplicons for *P. galeata* NIES-512 genomic DNA. In particular, mibC132, mibC196, and mibC300 showed only the expected target amplicons from the positive control, *P. galeata* NIES-512, whereas the others showed non-specific amplicons from both the positive and negative controls (Figure 2). These non-specific amplicons were not eliminated with various annealing temperatures of 58 °C‒64 °C. Only three primers, mibC132, mibC196, and mibC300, provided amplicons of the expected size without any additional amplicon when using both the positive and negative controls (Appendix A). Based on these results, the mibC132, mibC196, and mibC300 primers were applied to the environmental DNA samples collected from the sediment of the test stream (Gong-ji Stream). The experimental conditions for the PCR were the same as those employed in the previous test, resulting in clear, single, expected amplicons from both winter season (January and February) samples (Appendix A). This result confirmed that the primer sets designed in this study are capable of detecting genes related to 2-MIB synthesis without the production of any other non-specific PCR amplicon. In addition, amplicons of the expected size were synthesized using the *mibC* primers in five cyanobacterial strains isolated from the North Han River watershed without the production of any non-specific amplicon (Figure 3). In *P. mucicola*, none of the designed mibC primers amplified any amplicon (Figure 3). The mibC196 and mibC300 primers amplified the *mibC* gene in all five strains classified into Oscillatoriales and Synechococcales. However, the mibC132 primer only amplified the *mibC* gene from *P. raciborskii* (Figure 3c).

### 3.2. Phylogenic Analysis of Primer Amplicon Sequences

The selected primer sets, mibC132, mibC196, and mibC300, could detect *mibC* from both the cyanobacterial strain *P. galeata* NIES-512 and environmental samples. Thus, these primers could be used to evaluate the 2-MIB production potential and gene expression in environmental systems; however, this information is not sufficient to provide an early warning of 2-MIB occurrence. To achieve this goal, it is necessary to further identify strains that possess *mibC* in the cyanobacterial community. Accordingly, the sequences of amplicons generated using the mibC132, mibC196, and mibC300 primers were subjected to a phylogenetic analysis. The amplicon sequences were classified as being most closely related to *Pseudanabaena*, whereas they were phylogenetically separated from the *mibC* genes in *Oscillatoria*, *Leptolyngbya*, and *Planktothricoides*. As shown in Figure 3a, the sequences of mibC132 amplicons were phylogenetically more similar to those of *Pseudanabaena* and separate from those of *Oscillatoria*, *Planktothricoides,* and *Leptolyngbya*. However, the sequences amplified by the mibC196 and mibC300 primers were phylogenetically close to not only *Pseudanabaena*, including *Pseudanabaena* sp. NIVA-CYA 111 and *P. galeata* NIES-512, but also to *Oscillatoria* sp. and *Leptolyngbya* sp. (Appendix A). In addition, there was no difference in the sequences of amplicons from the field samples collected in Gong-ji Stream, demonstrating high phylogenetic similarity (Appendix A). All *mibC* genes amplified from five strains (except for *P. mucicola*) isolated from the North Han River were classified as the cyanobacterial *mibC* gene group, and the phylogenic branch was separated from that of the geosmin-synthesizing gene (*geoA*) of *Nostoc* (Figure 4). In particular, the *mibC* sequence of *L. redekei*, which does not exist in the GenBank database, was phylogenetically the most similar to that of *Pseudanabaena*, and was separated from those of *Planktothricoides*, *Lyngbya*, *Leptolyngbya*, and *Phormidium* (Figure 4). The *mibC* sequence of *Lyngbya* was the most similar to that of *Planktothricoides*, whereas the *Leptolyngbya* A2 strain isolated from China was separated into a different phylogenic branch from that comprising the North Han River strains. The *mibC* genes of *Phormidium* and *Leptolyngbya* showed a >90% identity with sequences of *Pseudanabaena* in BLAST analysis, although they were classified into different phylogenic branches.

Based on the phylogenetic analysis and electrophoresis of amplicons, the mibC196 and mibC300 primers showed relatively broad specificity toward Oscillatoriales and Synechococcales with 2-MIB production potential. By contrast, the mibC132 primer was more specific for detecting *Pseudanabaena* and *Planktothricoides,* which are known to produce the odorous 2-MIB compound [27,28,36].

### 3.3. Quantitative Analysis of In Vivo and In Vitro Samples

To achieve sufficient sensitivity for detecting *mibC* in the cyanobacterial community, we employed real-time PCR using *P. galeata* NIES-512 as the positive control strain. The correlation between the threshold cycle values in the real-time PCR analysis for each primer set and the *mibC* gene copy number, representing the amount of *mibC* gene, was fit to a linear regression, with R^2^ values for mibC132 and mibC196 of 0.999 and 0.998, respectively (Figure 5). The slopes of the standard curves of the primers mibC132 and mibC196 were −3.34 and −3.40, respectively, and their amplification efficiencies were 100–102% (Figure 5). Therefore, both primer sets could be applicable for real-time PCR without producing non-specific amplicons owing to primer self-binding (e.g., dimers).

To test the application of the mibC196 and mibC132 primers as useful molecular probes to detect *mibC* and to evaluate the 2-MIB production potential of cyanobacteria, we employed this primer set for detection with the field samples. The annual average concentration of 2-MIB from the water column of Gong-ji Stream collected between February of 2015 and February of 2016 was 17 ± 15 ng/L (0–49 ng/L), and the highest concentration was detected between August and November (Appendix A). Specifically, the highest 2-MIB concentration in the water column was 49 ng/L in September, followed by November at 45–49 ng/L. The average copy number of *mibC* from the water column and the sediment of Gong-ji Stream were 4.4 × 10^7^ (±14 × 10^7^) copies/mL and 9.8 × 10^4^ (±12.9 × 10^4^) copies/mL, respectively, and the temporal variation pattern of the copy number was similar to that of the 2-MIB concentration (Appendix A). In addition, the *mibC* expression pattern in the water column was very similar to the temporal fluctuations in *mibC* gene copy number and 2-MIB concentration (Appendix A).

The correlations between 2-MIB concentration and both *mibC* gene copy number and gene expression levels were statistically significant (*p* < 0.05; Figure 6a). The correlation coefficients between copy number and 2-MIB concentration in the sediment and water column were 0.749 and 0.644, respectively. The expression level of the *mibC* gene in the water column was also highly correlated with the 2-MIB concentration (r = 0.689). The copy number and expression level of *mibC* in the water column were significantly correlated (*p* > 0.01, r = 0.995) with the *P. limnetica* cell density (Figure 6b, Appendix A). By contrast, the *mibC* copy number in the sediment showed a low correlation with the cell density of *P. limnetica*, which was found in only the water column.

## 4. Discussion 

In this work, we developed and applied new molecular probes to detect *mibC,* which is related to the production of 2-MIB in cyanobacterial communities. Using the mibC132, mibC196, and mibC300 primers developed in this study, we were able to not only successfully detect the presence of the *mibC* gene from both single cyanobacterial strains and natural cyanobacterial assemblages with high specificity but could also quantify gene expression and environmental RNA with excellent amplification efficiency.

However, the other primers designed in this study (mibC117, mibC127, mibC200, mibC564, and mibC719) amplified non-specific amplicons during PCR. These unexpected non-specific amplicons could have been caused by non-specific binding [37], which could lead to overestimation of the 2-MIB production potential in quantitative analyses. Because the binding of primers to a target gene varies depending on the annealing temperature [37,38], we employed annealing temperatures of 55–64 °C for PCR against the positive control to eliminate noise. Under these conditions, only the mibC132, mibC196, and mibC300 primers provided amplicons of the expected size without any extra bands.

For primer design, it is critical to consider the G-C content, which strongly determines the melting temperature and PCR efficiency [39,40]; the G-C content of primers generally ranges from 40% to 60%, and repeated sequences should be excluded to avoid self-assembly [39]. In the present study, the five primer sets with a high G-C content and repeated sequences were excluded from the PCR tests. Moreover, biosynthesis of 2-MIB compound precursor is closely associated with photosynthetic pigment synthesis [41]. Therefore, the sequence of the target amplicon for mibC could be partially similar to those of genes related to chlorophyll-a synthesis, and it could result in the production of non-specific amplification, regardless of optimization of the annealing temperature.

The amplification efficiency is considered to be optimal when the amplification product size is 100–200 bp and the optimum efficiency (100%) is obtained when the slope of the standard curve is –3.3 in real-time PCR analysis [42,43]. The *mibC* primer previously reported to be used for quantification was larger than 200 bp, and amplification efficiencies were 97% and 95–102% with SYBR polymerase and TaqMan probe, respectively [14,18]. However, the *mibC* primers designed in this study produced an amplicon smaller than 200 bp, along with increased amplification efficiency of up to 100–102% using SYBR polymerase and an R^2^ value very close to one. Thus, we cautiously conclude that the mibC132, mibC196, and mibC300 primer sets are suitable to detect *mibC* in cyanobacterial communities and to evaluate the 2-MIB production potential and gene expression. As demonstrated in prior studies, 2-MIB is synthesized by a series of proteins encoded by the 2-MIB synthesis-associated operon [18,44], and it was suggested that the 2-MIB production potential could be assessed by detecting and quantifying genes in this operon, including *mibC* [14].

Our study also offers extension of applicability to estimate the in situ 2-MIB concentration and causative cyanobacterial density based on the relationships among *mibC* gene copy number, gene expression level, in situ amount of 2-MIB compound, and cyanobacterial density. However, quantitative prediction of these relationships requires further careful studies. In the quantitative analysis of *mibC* in the water column and sediment of the test site (Gong-ji Stream), the mibC196 primer results (copy number and gene expression level) showed a significant correlation with the actual in situ 2-MIB concentration and cell density of *P. limnetica*, which was demonstrated to produce 2-MIB in prior studies [27,45,46]. In particular, the gene expression level was highly correlated with the cell density of *P. limnetica* (r = 0.995, *p* < 0.01). Based on these results, *mibC* expression in the water column of Gong-ji Stream was suspected to be derived from *P. limnetica*. Therefore, the concentration of 2-MIB in the water column might originate from *P. limnetica* among the members of the diverse cyanobacterial community in the Gong-ji Stream.

## 5. Conclusions 

In this study, we developed the primers for detecting and quantifying 2-MIB-synthesizing cyanobacteria based on *mibC* gene sequences and investigated the 2-MIB production capability in 14 different cyanobacterial strains using these primer sets. Primer sets mibC196 and mibC300 showed universality to *mibC* in the Synechococcales and Oscillatoriales strains; the mibC132 primer showed high specificity for *Pseudanabaena* and *Planktothricoides mibC*. Our mibC primers also showed high correlations among related variables in the field (2-MIB concentration with water DNA and RNA and sediment DNA; cyanobacteria cell density with water RNA and DNA). We suggest that this approach can be applied to other freshwater systems, including drinking water resources in which off-flavor-related problems are caused by cyanobacterial blooms, and that the primers proposed as molecular probes could be used to provide information on the 2-MIB production potential and gene expression as an early warning monitoring system. This study, therefore, offers new primers with both high specificity for *mibC* gene detection and amplification efficiency of gene expression and is also the first study to evaluate the 2-MIB production potential and gene expression using a molecular biological method in freshwater systems in Korea. Lastly, we underline that this study provides findings to improve the conventional methods which have a limit to identify the presence of 2-MIB-producing cyanobacteria and to measure 2-MIB content in the sediment of aquatic systems.

## Figures and Tables

**Figure 1 ijerph-17-01933-f001:**
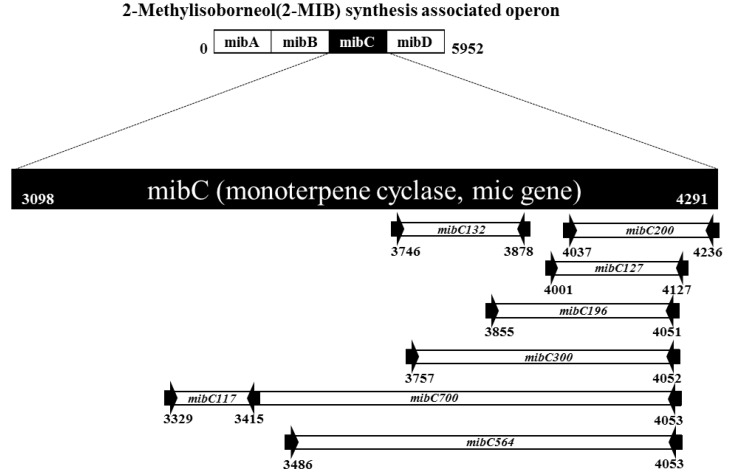
The 2-Methylisoborneol (2-MIB) synthesis-associated operon and positions of the designed primers on *mibC* (Adapted from Chiu et al., 2016). The sequences of *mibC* from the GenBank database, including *Pseudanabaena* sp. dqh15, were aligned, and the homologous region of *mibC* was selected for primer targeting. The arrows represent the directions of the primers, and the numbers in the primer names indicate the expected sizes (base pairs).

**Figure 2 ijerph-17-01933-f002:**
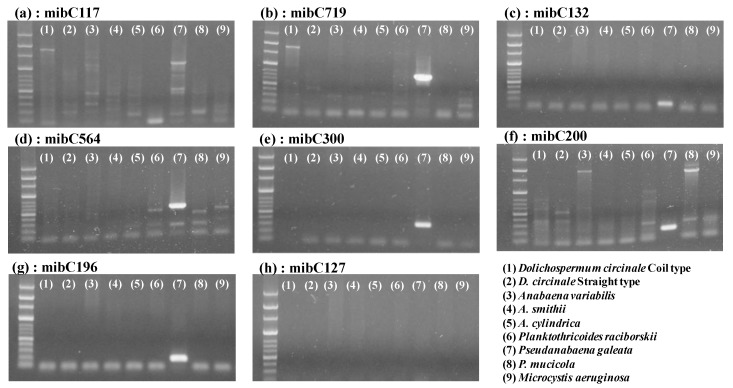
Amplification of the 2-MIB-synthesizing gene, *mibC*, by eight sets of primers using the genomic DNA of cyanobacteria strains cultured in the laboratory as templates. The PCR product generated by each primer set was analyzed by agarose gel electrophoresis. The electrophoresis gel shows the amplicons produced by PCR with (**a**) mibC117, (**b**) mibC719, (**c**) mibC132, (**d**) mibC564, (**e**) mibC300, (**f**) mibC200, (**g**) mibC196, and (**h**) mibC127. mibC127 did not produce amplicons.

**Figure 3 ijerph-17-01933-f003:**
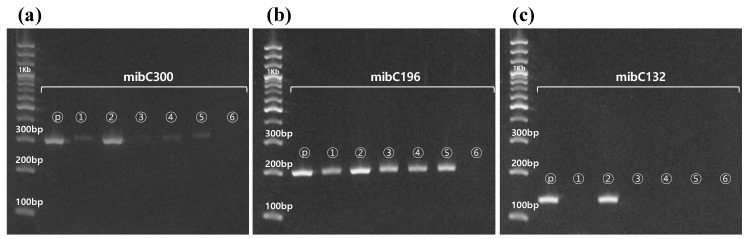
Amplification of the 2-MIB-synthesizing gene *mibC* by the primers mibC132, mibC196, and mibC300 using the genomic DNA of cyanobacteria strains isolated from the North Han River. The PCR product generated by each primer set was analyzed by agarose gel electrophoresis. The electrophoresis gel shows the amplicons produced by PCR with (**a**) mibC300 primer, (**b**) mibC196 primer, and (**c**) mibC132 primers. *P. mucicola* did not harbor *mibC*. Lane numbers in the figure represent *Pseudanabaena galeata* NIES-512 as a positive control (P), *Phormidium ambiguum* (1), *Planktothricoides raciborskii* (2), *Lyngbya murrayi* (3), *Leptolyngbya boryana* (4), *Limnothrix redekei* (5), and *Pseudanabaena mucicola* (6).

**Figure 4 ijerph-17-01933-f004:**
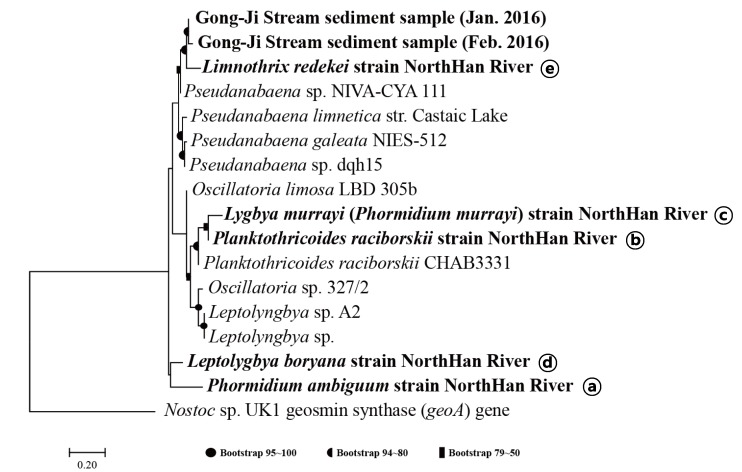
Phylogenic analysis of *mibC* sequences obtained using the mibC196 primer with sequences from cyanobacterial strains isolated from the North Han River and field sediment environmental DNA samples. The database DNA sequence was downloaded from NCBI GenBank, and the target gene sequence was selected by a keyword search (“*mibC*” or “MIB”). Phylogenic analysis of aligned sequences was performed using the maximum-likelihood method (1000 bootstrap replicates). *Nostoc* sp. UK1 strain was the root species and *P. mucicola* did not have the *mibC* gene.

**Figure 5 ijerph-17-01933-f005:**
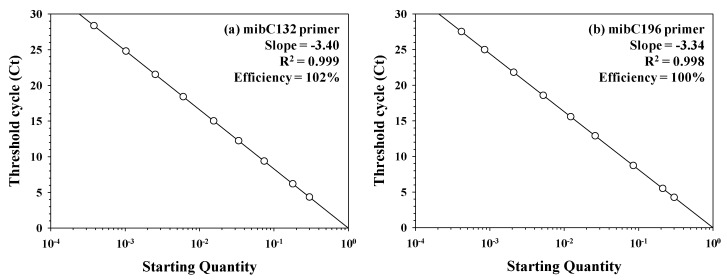
Standard curve for real-time PCR analysis with serial ten-fold-diluted DNA, which was cloned from the *mibC* gene fragment. Primer sets of (**a**) mibC132 and (**b**) mibC196 were used for SYBR type polymerase.

**Figure 6 ijerph-17-01933-f006:**
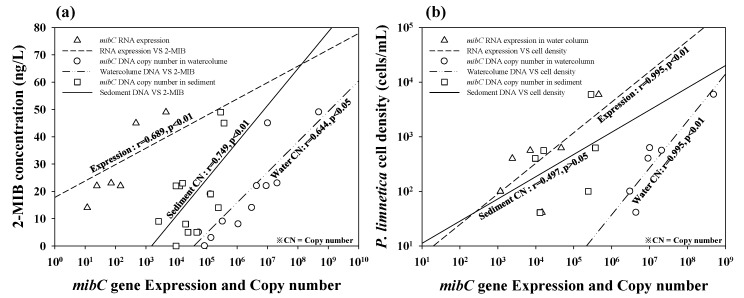
Regression lines for the correlation among mibC196 primer-derived amplicon copy number, gene expression, 2-MIB concentration, and *Pseudanabaena limnetica* cell density in the water column and sediment of Gong-ji Stream in 2015–2016. The 2-MIB compound concentration (**a**) and *P. limnetica* cell density (**b**) showed a linear relationship with the *mibC* gene amount (copy number and gene expression level). The amount of *mibC* was significantly correlated with both the 2-MIB concentration and *P. limnetica* cell density in the water column.

**Table 1 ijerph-17-01933-t001:** Sequences of the designed primers targeting *mibC* to evaluate 2-methylisoborneol (2-MIB) production potential in cyanobacteria.

No.	Name	Direction	Length (mer)	Sequence (5′ → 3′)	GC Contents (%)	Purpose	Amplicon Size (bp)	Melting Temp.
**1**	mibC117	Forward	19	CCG AGC GAT TCA ACG AGC C	63.2	Real-time PCR	117	60 °C
		Reverse	22	CCT CGT CCT CAT CAA ACC CGC A	59.1			
**2**	mibC127	Forward	20	CCC CTG TTA CGC CAC CTT CT	60.0	Real-time PCR	127	60 °C
		Reverse	24	TCA TGG AGG TGT AGA AGC TGT CGT	50.0			
**3**	mibC132	Forward	22	CGY ACC TGT TAC GCC ACC TTC T	56.8	Real-time PCR	132	60 °C
		Reverse	24	TCA TGG AGG TGT AGA AGC TGT CGT	50.0			
**4**	mibC196	Forward	24	ACG ACA GCT TCT ACA CCT CCA TGA	50.0	Real-time PCR Screening	196	64 °C
		Reverse	26	AAT CTG TAG CAC CAT GTT GAC WGG TG	46.2			
**5**	mibC200	Forward	24	ACA TGG TGC TAC AGA TTG CGG CGG	58.3	Real-time PCR Screening	200	64 °C
		Reverse	24	GCT GTT ATG CCA TTC AAA TGC GCC	50.0			
**6**	mibC300	Forward	23	TGT TAC GCC ACC TTC TCT ATG TT	43.5	Screening	300	64 °C
		Reverse	21	CAA TCT GTA GCA CCA TGT TGA	42.9			
**7**	mibC564	Forward	22	TCG GCT GTT GAT YGG KGC CAA G	59.1	Screening	564	64 °C
		Reverse	25	AAT CTG TAG CAC CAT GTT GAC WGG T	44.0			
**8**	mibC719	Forward	19	CCG AGC GAT TCA ACG AGC C	63.2	Screening	719	60 °C
		Reverse	25	AAT CTG TAG CAC CAT GTT GAC WGG T	44.0			

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
