# Peer review of "Molecular Probes to Evaluate the Synthesis and Production Potential of an Odorous Compound (2-methylisoborneol) in Cyanobacteria"

_ijerph, 2020, doi:10.3390/ijerph17061933_

Round 1
Reviewer 1 Report
Undesirable taste and odor of drinking and recreational waters can indicate quality problems or possible risks for human health. One example of this situation is the presence of cyanobacterial metabolites in water. The manuscript entitled “Molecular probes to evaluate the synthesis and production potential of an odorous compound (2-methylisoborneol) in cyanobacteria” could find a wide audience. In many countries, the problem of unpleasant drinking water smell is a tangible issue. In spite of all, easy-to-use, sensitive methods for the detection of causative agents are still not available. In my opinion, the methodological part of the manuscript is poorly described, and some analyzes were incorrectly performed. In this case it is difficult to accept the manuscript for publication in the present form. I did not evaluate the discussion part, since M&M and Results sections contain serious weaknesses.
Some specific comments:
Introduction:
we do not have strains in the environment, a strain is made up of the descendants of a single isolation in pure culture (in reference to “cyanobacterial community in freshwater systems is composed of a wide variety of strains”) “There has been increased research interest in factors related to 2-MIB production in recent decade” – what does it mean? Factors regulating 2-MIB production?
Materials and methods:
1. Anabaena cylindrica not “cylindrical’; What’s the purpose of 3-month cyanobacteria culturing and media exchange? 2. “In brief, the cells of each strain were mixed with lysis solution and
silica beads, and then the mixture was vortexed vigorously. The supernatant was applied to the resin in the micro-column of the DNA extraction kit” – not needed; instead, there is nothing on RNA isolation, RT conditions
3. as I understood the Authors used MEGA to align the sequences, but it is not explained how did they design the primers 4. the Authors mention in this section that they tested annealing temperature within the range of 60-64, in Fig. S2 we can see other values (58-64)
Results:
Which annealing temperature was chosen for PCR? Was it really impossible to optimize the reaction for e.g. mibC719 primers? The product has reasonable size and is clearly visible in the positive control? For e.g. mibC132 and 196 (smaller products) other ladder and electrophoresis conditions should have been chosen! Sometimes it’s confusing in the text when the Authors write “primer mibCXXX” while it is a set of primers (forward and reverse) In M&M the Authors wrote that they sampled in 2015, in the Results we also have year 2016 2 – why mibC196 results are presented two times and there is no result for P. musicola, it should be there, even though the product was not detected 2. and Fig. 3: no bootstrap support values, I cannot draw any conclusions, no explanation what stands for “in vivo positive control” 4 You cannot align sequences from different genes! (geoA, mibC), no bootstrap values, again 5 and 3.3. – it’s hard to evaluate this part; In the text the Authors wrote that they used P. galeata DNA, in the figure captions they mention “plasmid” ? The x and y axes are incorrectly described. How the Authors measured MIB concentration?

Author Response
- The review was very helpful. All the comments were very constructive, and we agree with all the suggestions. Particularly, we recognize that there were some faults in the manuscript including wrong or awkward expression and some poor description of methodology. We are very grateful to the reviewer to point out them and to provide comments to correct problems. We revised the manuscript according to every piece of the reviewer comments with some additional description of methods. Finally, the revised manuscript was gone through English language editing by a professional native editor (Editage: www.editage.co.kr).
- Detailed responses are attached in a separated file (Words).

Reviewer 2 Report
This manuscript needs major revision before it could be considered for publication. The following comments should be considered:
1.Put the exact coordinates of the location in Gong-ji Stream. Which part of the estuary region did you collect the samples?
2. Discussion in figure legends should be avoided. Write your discussion on discussion of results.
3. References should be according to format. I am not sure, why Korean and Japanese languages are present here.
4. You have so many Phylogenic trees, so better make the flow of your paper organized.
5. Figure 5, why x axis is labelled as starting quantity? strange
Author Response
- The review was very helpful. We are very grateful to the reviewer comments. All the comments were very constructive, and we agree with all the suggestions. We considered them all carefully and revised the manuscript according to the reviewer comments. Finally, the revised manuscript was gone through English language editing by a professional native editor (Editage: www.editage.co.kr).
- Detailed responses are presented in a separated file (Words)

Reviewer 3 Report
General comment: In this study new molecular probes were developed and tested to detect mibC, which is related to the production of 2-MIB in cyanobacterial communities. Using the primers mibC132, mibC196, and mibC300 developed the authors were able not only to detect wth success the presence of the mibC gene from both single cyanobacterial strains and natural cyanobacterial assemblages with high specificity but could also quantify gene expression in environmental RNA with excellent amplification efficiency. I think this work is very useful in the subject of environmental molecular analysis of potential production of an odorous compound (2-methylisoborneol) in cyanobacteria. Regarding the often appearance of cyanobacteria blooms in waters, this kind of studies is very important to detect pollution grades originated by volatile compounds.
Specific comments:
Introduction:
Maybe an introductory sentence is needed in the begining of the introduction. I don’t like the way it begins immediatelly with: “Terpenoid metabolites”
Methods:
Pag. 2: “During medium changes, the dead cells and cells in death phase which were settled down onto the bottom of flask and paled into yellow-green color, were removed along with the medium.”
What do the authors mean with death cells and cells in death phase? What is the difference?
How do they removed dead cells? How do they guarantee that they were really dead? Through microscopic analysis?
Pag. 3: “For genomic DNA extraction, cyanobacterial strains (P. galeata, P. mucicola, O. redekei [L. redekei], and P. raciborskii NO) attached to the bottom of flasks were collected using a scraper, whereas the other strains (M. aeruginosa, D. circinale ST, D. circinale CT, A. variabilis, A. smithii, A. cylindrical, L. murrayi, P. ambiguum, L. boryana, and P. raciborskii strain PO) were harvested by filtration of 10 mL of culture using a polycarbonate filter (ø25 mm, pore: 1 μm, Whatman, USA).”
How can authors be sure that these species are really down and the others floating? With such a precise identification?
The results and discussion are very well written and organized.
Author Response
- The review was very helpful. All the comments were very constructive, and we agree with all the suggestions. We are very grateful to the reviewer comments. We considered them all carefully and revised the manuscript according to the comments. Finally, the revised manuscript was gone through English language editing by a professional native editor (Editage: www.editage.co.kr).
- Detailed responses are presented in a separated file (Words).

Round 2
Reviewer 1 Report
Dear Authors,
I appreciate the changes made in the manuscript. However, there are some points that cause my concerns:
The abstract:
- Sequences are being deposited in the GenBank, not archived.
- You cannot collect cyanobacterial strains from the environment, you can isolate them
- I guess it's not that the stream has high 2-MIB concentration, but high 2-MIB concentrations are/were detected in the waters samples from that stream
- The sentence about the amplification efficiency is strange, plus r=689?
Introduction:
- In reference to my previous comment: I pointed out that the term "strain" refers to laboratory, not the environment. In the manuscript, wouldn't it be better to talk about species and than genotypes? Because different species can produce these odorous metabolites and than if we are more specific, the production may differ at intraspecies level.
M&M:
- 2.2. - the tittle should be changed, the Authors added paragraph on RNA extraction
- The sentence "The genomic DNA of the algal strain and the field water column...." sounds strange. Additionally is only one example but refers also to other parts of the manuscript where the Authors use singular from rather than plural.
- 2.2. The Authors do not mention that they isolated DNA from sediments
- 2.2. - last sentence, RNA concentration was not measured?
- 2.4. It should be described how the Authors cleaned the cut bands.
- 2.4. The information on RT-qPCR is insufficient
- 2.4. There is no information on polymerase, primers concentration etc (PCR for mibC)
Results:
- 3.2. The Authors refer to mibC presence in the environemntal samples, but in the previous section we can only find the results concerning cyanobacterial strains
- My question is why the Authors did not deposited the sequences in the GenBank and why they searched the homologous sequences using "key word search", not BLASTn?
- 3.2. As pointed by the Authors, of course you can root the tree, add the outgroup, but the sequence of the same gene should be used! It looks like the Authors use the sequences of two different genes, geoA and mibC, interchangeably for different strains. I can understand that after some reference revision (e.g. https://www.ncbi.nlm.nih.gov/pubmed/25462716), but for the random Reader it's unclear.
- I would improve the trees, put only the Species/genera name, strain code and GenBank accession number. It's a common practice.
Discussion:
- I do not understand the sentence in the third paragraph starting with "Moreover"
- I would more clearly underline why the obtained results are important and worth publishing.
Author Response
Response to reviewer comments (Round 2)
Reviewer 1
- We are very grateful to the reviewer’s constructive comments again. We considered them all carefully and revised the manuscript according to the comments.
Dear Authors,
I appreciate the changes made in the manuscript. However, there are some points that cause my concerns:
The abstract:
- Sequences are being deposited in the GenBank, not archived.
- According to the comment, we replaced the word ‘archived’ with ‘deposited’ in the abstract.
- You cannot collect cyanobacterial strains from the environment, you can isolate them.
- According to the comment, we replaced the word ‘strain’ with ‘cells’ in the abstract.
- I guess it's not that the stream has high 2-MIB concentration, but high 2-MIB concentrations are/were detected in the waters samples from that stream.
- The Gong-ji stream is one of the major streams which flow into the Uiam reservoir through the watershed of a big city nearby. As the study site is the estuary region of stream, stream water flows very slowly towards the estuary, and the stream mouth region has high concentration of nutrients. Also, affected by the reservoir (dam), water often flows backward to the stream. For this reason, algal blooms occur frequently in the estuary region of stream, with resultant high 2-MIB concentration.
- The sentence about the amplification efficiency is strange, plus r=689?
- We modified awkward expression according to the reviewer comment, as following: “Our mibC primers showed excellent amplification efficiency (100-102%) and high correlations among related variables (2-MIB concentration with water RNA r = 689, p < 0.01; sediment DNA r = 0.794, p < 0.01; and water DNA r = 0.644, p < 0.05; cyanobacteria cell density with water RNA and DNA r = 0.995, p < 0.01).”
Introduction:
- In reference to my previous comment: I pointed out that the term "strain" refers to laboratory, not the environment. In the manuscript, wouldn't it be better to talk about species and than genotypes? Because different species can produce these odorous metabolites and than if we are more specific, the production may differ at intraspecies level.
- According to reviewer comment, we replaced the word ‘genotype’ with ‘species’ in the manuscript.
Materials and methods:
- 2.2. - the tittle should be changed, the Authors added paragraph on RNA extraction.
- According to reviewer comment, we changed the M&M 2.2 section title and rearranged to separate the corresponding paragraph.
- The sentence "The genomic DNA of the algal strain and the field water column...." sounds strange. Additionally is only one example but refers also to other parts of the manuscript where the Authors use singular from rather than plural.
- According to reviewer comment, we modified the corresponding sentence in M&M 2.2 section as following: “The genomic DNA of the algal strain was extracted using a DNA extraction kit for soil (Macherey-Nagal, Germany) according to the manufacturer’s instructions, after cell lysis via the physical bead beating method. Both water column and sediment samples were extracted DNA using the same method as for algal strain DNA extraction.”
- In addition, we rearranged to separate RNA part in 2.2 section as indicated in above reply.
- 2.2. The Authors do not mention that they isolated DNA from sediments.
- We appreciate the reviewer’s pointing out about this. We used the same extraction method as for algal strain. We added a new sentence as following: “Both water column and sediment samples were extracted DNA using the same method as for algal strain DNA extraction.”
- 2.2. - last sentence, RNA concentration was not measured?
- We appreciate this comment. Yes, we measured RNA concentration by the same equipment as for DNA concentration. We also added ‘RNA’ in the last sentence of M&M 2.2 section.
- 2.4. It should be described how the Authors cleaned the cut bands.
- We added detailed description about process of agarose gel extraction in M&M 2.4 section as following: “The bands of the agarose gel were cut by a surgical scalpel after approximating each size (ca. 5mm x 3mm) by a ruler. From the middle of agarose gel band, about 2mm of band edge was removed. The PCR product was extracted from the band cut using a Gel & PCR purification system (HiGeneTM, GP104-200, Biofact Co., Korea).”
- 2.4. The information on RT-qPCR is insufficient and there is no information on polymerase, primers concentration etc (PCR for mibC).
- According to the reviewer comment, we added detailed information about real-time PCR performance (polymerase, acquiring fluorescence, etc..) in M&M 2.4 section, as following: “In addition, real-time PCR was performed using the 2× Real-Time PCR Master Mix: including SFCgreen®I in mixture (DQ362-40h, Biofact Co. Korea) in a real-time PCR cycler (Rotorgene, Qiagen Co., Germany)….”
- We added information of SFCgreen fluorescence value as following: “The SFCgreen fluorescence value was obtained in each cycle of the extension step.”
- We also added detailed information about mixture recipe for real-time PCR analysis in M&M 2.4 section as following: “Each template of DNA and RNA was mixed with polymerase mixture for the real-time PCR analysis. The mixture was blended with all necessary materials, including 2X polymerase master mix (SFCgreen-intercalate type) 10 μL, forward primer (10 pmole/ μL) 1 μL, reverse primer (10 pmole/ μL) 1 μL, template 5 μL, and Nuclease free water 3 μL.”
Results:
- 3.2. The Authors refer to mibC presence in the environemntal samples, but in the previous section we can only find the results concerning cyanobacterial strains.
- We provided the electrophoresis results of environmental samples in ‘Supplementary Figure 3’. And we already described it for the supplementary data in Results 3.1 section.
- My question is why the Authors did not deposited the sequences in the GenBank and why they searched the homologous sequences using "key word search", not BLASTn?
- We appreciate this suggestion. We will submit mibC sequences to DNA database after we publish this paper for the reference of them.
- In this study, we designed the primers first; and then we detected mibC gene of cyanobacterial strains by using the designed primers. For this reason, we had to design a primer sequence based on mibC gene sequences data which were deposited in NCBI Genbank. As we did not have a mibC sequence of the strains before designing the primers, we could not use BLASTn. So, we used keyword search to collect the mibC gene sequence.
- 3.2. As pointed by the Authors, of course you can root the tree, add the outgroup, but the sequence of the same gene should be used! It looks like the Authors use the sequences of two different genes, geoA and mibC, interchangeably for different strains. I can understand that after some reference revision (e.g. ttps://www.ncbi.nlm.nih.gov/pubmed/25462716), but for the random Reader it's unclear.
- We appreciate the reviewer instructive comment.
- I would improve the trees, put only the Species/genera name, strain code and GenBank accession number. It's a common practice.
- According to reviewer comment, we omitted the names of functional gene in Figure 4.
Discussion:
- I do not understand the sentence in the third paragraph starting with "Moreover".
- Both pathways of 2-MIB and chlorophyll-a synthesis are very similar, and they use the same precursor in the synthesis of 2-MIB and carotenoid pigment. Therefore, we assumed that these two functional gene sequences related to biosynthesis of 2-MIB and chlorophyll-a might be similar each other. For this reason, we suspected that some mibC primers (e.g., mibC117, mibC564, and mibC200 primer, etc.) produced many noise bands.
- We modified the part of paragraph starting with ‘Moreover’ according to reviewer comment as following: “Moreover, biosynthesis of 2-MIB compound precursor is closely associated with photosynthetic pigment synthesis. Therefore, the sequence of the target amplicon for mibC could be partially similar to those of genes related to chlorophyll-a synthesis, and it could result in the production of non-specific amplification, regardless of optimization of the annealing temperature.”
- I would more clearly underline why the obtained results are important and worth publishing.
- We are very grateful to this comment. As we reviewed the manuscript, especially the discussion section, there we already put the significance of our work and findings. However, we added a sentence in the last of discussion to underscore applicability of our method and new primers to the sediment of freshwater, which is hard to study by the conventional methods, as following: “Lastly, we underline that this study provides findings to improve the conventional methods which have a limit to identify the presence of 2-MIB-producing cyanobacteria and to measure 2-MIB content in the sediment of aquatic system
End of author response.

Reviewer 2 Report
OK
Author Response
Dear reviewer:
We are very grateful to you providing us acceptable review result to our first revised manuscript. Since you have not given us any specific suggestions on the revision, we cannot provide a detailed Author Reply either. However, we've modified some parts of the revised manuscript according to the other reviewer comments. You can find our work in the second revision.
Thank you very much again for your constructive review comments on our manuscript, and with putting them in the revision, we believe that the manuscript is now much improved.
Sincerely,
Soon-Jin Hwang
Corresponding author